# Peer review of "Arginase Isoform Expression in Chronic Rhinosinusitis"

_jcm, 2019, doi:10.3390/jcm8111809_

Round 1

Reviewer 1 Report

This is an interesting preliminary paper showing that Arginase 11(ARG2) is increased in certain forms of chronic rhinosinusitis, predominantly the non - T2 variety, although the authors do not go so far as to elucidate this observation I think they should put it forward.

The Han classification does not appear to distinguish polyp from non- polyp CRS, whereas the authors have sensibly also looked at this distinction and found that CRS with polyps does not differ significantly from controls with respect to ARG2.This agrees with the previous observations of Taruya ( Ref 4).

It is known (Colantonio et al, ref.48) that nNO corresponds to the grade of nasal polyposis- did the authors find that ARG 2 levels varied with the extent of polyps?

The numbers of patients, n=3 and n= 5 , in some of the Han groups are small, therefore more work needs to be done before lines 256-264 can be included.The authors could also look at the effects of therapy of various kinds on ARG 2 levels, as was done for NO. ( Ragab SM et al. Allergy 2006, 6, 717-724).

The figures should report the nature of the error bars- are they standard errors or standard deviation?

 Was it possible to undertake duplicate analyses of one biopsy? If so what was the repeatability?

There are several minor errors :citrulline is wrongly spelled throughout, there should be no apostrophe after Its( lines 173 & 279), wich (line 167), should be which,   fist ( line 157 ) should be first, defence ( line158) is the UK spelling.

Reviewer 2 Report

Diana Vlad and Silviu Albu wished to investigate the activity of arginase I (ARG1) and II (ARG2) in CRS. They found significantly higher ARG2 values In patients with chronic rhinosinusitis, compared to the control group. This is the only consistent result.

Critical points

Authors report ARG2 values observed in subgroups (with or without polyps, with or without asthma, atopic or non atopic ) compared to controls. They should analyze whether ARG2 values differed among the subgroups of patients with CRS. In other words, they should analyze the effect of atopy, polyps, asthma comorbidity on the ARG2 activity.I suspect that there are no difference among CRS subgroups. In discussion Authors do not discuss their results but they report on the physiological role of NO and its role in the pathogenesis of CRS.

Round 2

Reviewer 2 Report

The manuscript has been revised according to criticisms.